# Solid Anorganic Particles and Chronic Rhinosinusitis: A Histopathology Study

**DOI:** 10.3390/ijerph19127269

**Published:** 2022-06-14

**Authors:** Lenka Čábalová, Kristina Čabanová, Hana Bielniková, Jana Kukutschová, Jana Dvořáčková, Karol Zeleník, Pavel Komínek

**Affiliations:** 1Department of Otorhinolaryngology, Head and Neck Surgery, University Hospital Ostrava, 708 52 Ostrava, Czech Republic; lenka.cabalova@fno.cz (L.Č.); pavel.kominek@fno.cz (P.K.); 2Centre for Advanced Innovation Technologies, VŠ—Technical University of Ostrava, 708 00 Ostrava, Czech Republic; kristina.cabanova@vsb.cz (K.Č.); jana.kukutschova@vsb.cz (J.K.); 3Faculty of Mining and Geology, VŠB—Technical University of Ostrava, 708 00 Ostrava, Czech Republic; 4Institute of Pathology, University Hospital Ostrava, 708 52 Ostrava, Czech Republic; hana.bielnikova@fno.cz (H.B.); jana.dvorackova@fno.cz (J.D.); 5Faculty of Medicine, University of Ostrava, 703 00 Ostrava, Czech Republic

**Keywords:** fine and ultrafine particles, micro- and nanosized particles, airborne pollutants, nanotoxicology, chronic rhinosinusitis, Raman microspectroscopy

## Abstract

Although extensive research has shown the pathological effect of fine and ultrafine airborne particles, clear evidence of association of environmental exposure to them and inflammatory changes in human nasal mucosa is missing. Meanwhile, pathogenesis of chronic rhinosinusitis, despite being a disease with high prevalence in the population, is still unclear. The increasing evidence of the pro-inflammatory properties of these particles raises the question of their possible role in chronic rhinosinusitis. The presented study focused on detection of microsized anorganic particles and clusters of nanosized anorganic particles in the nasal mucosa of patients with chronic rhinosinusitis by Raman microspectroscopy and comparison of their composition to histologic findings. The results were compared to the findings in mucosa obtained from cadavers with no history of chronic rhinosinusitis. Solid particles were found in 90% of tissue samples in the group with chronic rhinosinusitis, showing histologic signs of inflammation in 95%, while in the control group, the particles were found in 20% of samples, with normal histologic findings in all of them. The main detected compounds were graphite, TiO_2_, amorphous carbon, calcite, ankerite and iron compounds. The results are in accordance with the premise that exogenous airborne particles interact with the nasal mucosa and possibly deposit in it in cases where the epithelial barrier is compromised in chronic rhinosinusitis.

## 1. Introduction

### 1.1. Definition and Difference between Fine and Ultrafine (Micro- and Nanosized) Particles

Particulate matter (PM) pollution is one a threat to human health. The particles of pollutants vary chemically and in size, depending on the sources from which they are emitted. These PM are divided into individual fractions according to their aerodynamic diameter, namely coarse (2.5–10 μm), fine (0.1–2.5 μm) and ultrafine (<0.1 μm) [1,2]. 

Ultrafine particles (UFPs), also referred to as PM_0.1_, are aerosols with an aerodynamic diameter below 0.1 µm (100 nm). These ultrafine particles are present in large amounts in the air, which means they pose a greater health risk. They usually enter the body through inhalation or ingestion [3]. 

The terms “fine” and “ultrafine particles” are reserved to airborne particles; therefore, when referring to particles in tissues, the terms “micro-” and “nanosized particles” (MPs and NPs) should rather be used, while the cut-off size between micro- and nano- remains the same, i.e., 100 nm.

Due to their extremely small scale, the relative surface of NPs is larger and thus the number of reactive atom groups on their surface is several times higher than in MPs and larger particles [4]. They also exhibit increased diffusivity and delayed sedimentation—therefore, they tend to stay airborne in the atmosphere for longer periods of time than MPs, which increases the risk of their inhalation [5,6]. They are ubiquitous in the environment, composed mostly of metal oxides and carbon, predominantly of antropogenous origin—produced, for example, by smoking, fossil fuel combustion, welding or road traffic [7,8]. 

They have been proven to have pathological effect on living cells by numerous in vitro studies and on animal models [6,9,10,11]. They can penetrate tissues both paracellularly and through cells and become bloodborne, redistribute in the organism and accumulate in vital organs, while the possibility of their lysosomal degradation, in contrast to MPs, is limited [7]. Due to their size being similar to the size of subcellular structures and increased reactivity, they can interfere with the function of organelles, enzymes or antibodies and also exhibit genotoxic properties, by induction of oxidative stress and by direct interaction with DNA [10,12,13].

All the above being said, it is quite rare to detect individual NPs in the tissues because they tend to form clusters probably due to the interaction with living cells and possibly due to processing of the tissue samples as well. For example, titanium dioxide, vastly used in various products in its nanosized form, probably enters the living organisms as a pollutant predominantly in the form of NPs; however, in tissues, it has been found mostly in the form of clusters [14,15]. 

### 1.2. Possible Role of Solid Anorganic Particles in Chronic Rhinosinusitis

The respiratory tract is, aside from the skin and the digestive tract, one of the main routes of entrance of solid anorganic particles into the organism [16]. The nasal cavity is the first passage through which air enters during respiration and several numerical models suggest that it is also the region where both MPs and NPs deposit the most out of the whole respiratory tract [17,18,19]. According to the “European Position Paper on Rhinosinusitis and Nasal Polyps 2020”, chronic rhinosinusitis (CRS) in adults is defined as the presence of two or more symptoms, one of which should be either nasal blockage/obstruction/congestion or nasal discharge (anterior/posterior nasal drip); or facial pain/pressure and/or reduction of loss of smell for ≥12 weeks [20]. Although CRS has a high prevalence in the population (estimated at 5.5–28%) and represents a significant burden not only to the quality of life of individual patients but is also undoubtedly a financially burdensome disease (for example, the estimated overall economic cost is USD 20–30 billion annually in the USA alone), its etiology and pathogenesis is still unclear, despite extensive research [20,21]. The emerging view is that CRS is a syndrome with multifactorial etiology in which interaction between environmental factors and host immune system play crucial roles [20]. Although evidence suggests that sinonasal epithelial barrier disruption and mucociliary clearance dysfunction leading to chronic mucosal inflammation may be triggered by environmental pollution, data on the specific pollutants, the synergy of their actions and the mechanism of their role in CRS pathogenesis is still insufficient [21,22].

The increasing evidence of cytotoxic and pro-inflammatory properties of airborne MPs and NPs naturally raises the question of their possible role in chronic inflammatory diseases, such as CRS.

### 1.3. Clinical Research of MPs and NPs and the Aim of the Study

Although it is clear that airborne pollutants pose a health, occupational and environmental hazard, only a few clinical studies focusing on micro- and nanosized particles have been conducted so far—for example, Zeleník et al. studied the presence of NPs in tonsillar tissue and Munger et al. analyzed metabolic, hematologic and urinalysis measures in healthy volunteers after administration of silver nanoparticles [8,23]. However, clear evidence of the association of exposure to MPs and NPs present in the living environment as air pollutants and inflammatory changes in nasal mucosa is still missing. 

The presented study follows up on the authors’ pilot study focused on the possibility of detection and evaluation of MPs and NPs in nasal mucosa obtained by endoscopic mucotomy in patients diagnosed with CRS by Raman microspectroscopy. A novel method of identification of said particles in tissue samples was developed and applied on a limited number of samples [16,24]. In the following study, the authors present a new set of results obtained by this Raman microspectroscopy mapping; in total, using 40 tissue samples obtained from patients with CRS and 10 reference samples, we discuss the possible relationship between the compounds found in the samples and inflammatory processes and compare the results with histological findings, smoking and occupational status [24].

## 2. Materials and Methods

This prospective study was conducted from September 2013 to March 2015, and was approved by the Institutional Ethics Committee of University Hospital Ostrava (identifier FNO-ENT-Nanoparticles, 2 RVO-FNOs/2013) and registered at ClinicalTrials.gov (identifier NCT02270125). The study was performed in accordance with the Declaration of Helsinki, good clinical practice, and applicable regulatory requirements. Informed consent was obtained from all participants before initiation of any procedure.

The inclusion criteria for the study group (CRS group) were: -Age over 18 years;-Clinical diagnosis of CRS according to EPOS2012 criteria (the criteria are identical to criteria named in EPOS2020);-Endoscopically verified hypertrophy of inferior turbinates;-Insufficient response to conservative therapy (nasal corticosteroid spray administration for 6 months or longer), as subjectively assessed by the patient;-No history of turbinate surgery and/or sinonasal tumor;-Immunocompetency and aptitude for surgery under general anaesthesia;-Written informed consent obtained.

A randomly selected group of 40 patients meeting the inclusion criteria, aged 20–78 (mean 41.3), 31 males and 9 females, were enrolled in the study. Their demographic and occupational history and history of smoking was obtained (Table 1).

The inclusion criteria for the control group (cadaveric donors) were: -Age over 18 years.-No history of chronic inflammatory disease of sinonasal mucosa (i.e., CRS, allergic fungal rhinosinusitis, allergic rhinitis) or other significant pathologies and surgical procedures in sinonasal region during their life (for example sinonasal tumors);-No endoscopic signs of inferior turbinate hypertrophy.

A total of 10 samples were obtained from randomly selected cadaveric donors meeting the inclusion criteria, aged 37–87 (mean 69), 6 males and 4 females (Table 2). 

All the subjects from both groups shared similar living environment regarding airborne pollution levels—all of them resided in the same region (Moravian–Silesian region, Czech Republic). 

### 2.1. Sample Acquisition and Preparation

The tissue samples were obtained by endoscopic “cold-steel” mucotomy under general anesthesia. The same technique was used in case of cadaver samples. The samples were attached to paraffin tablets by sterile surgical needles, immersed in 10% formalin and sent to the Institute of Pathology under sterile conditions for further processing. After alcohol–xylene dehydration and automated paraffin embedding, 2–4 μm thin sections were cut and mounted onto glass microscope slides, before staining with hematoxylin/eosin for routine pathological examination (Table 1 and Table 2). 

### 2.2. Raman Microspectroscopy

Raman spectra allowing chemical characterization of particles/clusters in the samples were obtained using Smart Raman Microscopy System XploRA (HORIBA Jobin Yvon, France). Raman spectra were acquired in the whole range from 100 to 4000 cm^−1^. A 100× objective lens was used, which is more suitable for the samples in thin films or fluids dried on a glass slide. The 532 nm excitation laser source (20–25 mW) with the laser spot diameter of approximately 0.5 µm allowing point analysis of particles/clusters was used. The intensity of the laser was regulated with regard to the measured sample—a lower intensity of the laser beam was set due to the potential of damage to organic samples. XploRA device allows 0.1, 1, 10, 25, 50 and 100% intensity of the initial laser beam. The acquisition time and the number of accumulations were set according to each sample to reduce the signal/noise ratio. Grating with 1200 grooves/mm was set. A screen image recorder camera attached to the microscope enabled the visualization of white-light images and the selection of the area of interest. In addition, the recording of the spectral images (Raman mapping) was performed in a selected region with 1 µm step. Measured Raman spectra were corrected using the LabSpec 5 software (HORIBA, Grenoble, France). The baseline correction (polynomial, order 8th), smoothing (Linear Savitsky-Golay filter, 2nd order, 9 points), and marking the position of the Raman bands in the measured individual Raman spectra were performed by LabSpec 5 software (HORIBA, Grenoble, France) [24].

### 2.3. Histological Examination

All the samples underwent routine pathological examination which was performed by a single pathologist in all the cases. The histological signs of severity of inflammation were subjectively visually assessed (Table 1 and Table 2): 0Normal histology (no inflammation);1Epithelial hyperplasia;2Epithelial hyperplasia with mild signs of inflammation;3Chronic inflammation.

“Epitheal hyperplasia“ was defined as presence of hyperplastic cylindrical epithelium with thickened basement membrane. Mild signs of inflammation in the submucosal space were characterized as presence of “focal”, “discrete” or “minimal” inflammatory cellularization (lymphoplasmocytic cellularization with occasional presence of neutrophiles and eosinophiles). Chronic inflammation was defined as massive presence of inflammatory cellularization (presence of large clusters of lymphocytes with presence of plasmocytes, macrophages and fibroblasts, and/or overall massive inflammatory infiltration of the tissue, i.e., not only focal, as subjectively assessed by the same pathologist). 

### 2.4. Correlation of Detected Compounds, Histology and Smoking and Occupational History

For the purpose of correlation of histological findings with the Raman mapping findings, the CRS group was divided into two subgroups: group A (histological signs of severity of inflammation 0–2, i.e., no or mild inflammation) and group B (histological signs of severity of inflammation 3, i.e., severe inflammation). The presence and composition of detected compounds were compared in the two groups. 

According to the smoking and occupational history, CRS and control groups were divided to smoker/non-smoker subgroups and “manual” workers (M)/“office” workers (O) subgroups. M/O subgroups were divided according to their estimated risk of occupational exposure to airborne pollution (M—high risk: for example, welder, policeman, tinsmith, etc.; O—low risk: for example, manager, programmer, student, etc.). 

## 3. Results

### 3.1. Detected Compounds

Solid anorganic MPs and clusters of NPs of average size of 0.5 µm were found in 90% (36/40) tissue samples of the CRS group and in 20% (2/10) tissue samples of the control group. 

The main detected compounds in the CRS group tissue samples were graphite, TiO_2_ (anatase, rutile), amorphous carbon, CaCO_3_ (calcite), Ca(Fe, Mg, Mn)(CO_3_)_2_ (ankerite) and iron compounds Fe_3_O_4_ (magnetite) and Fe_2_O_3_ (hematite)—the results are summarized in Table 3. 

BaSO_4_ (barite), Al (aluminium) and Si (silicon) were detected sporadically.

In the control group, only TiO_2_ (anatase) and amorphous carbon were detected, each in one sample (Table 2). 

Examples of Raman spectra are provided in Figure 1, Figure 2, Figure 3, Figure 4 and Figure 5. 

### 3.2. Histology

Histological signs of severity of inflammation in the CRS group were assessed subjectively by a single pathologist (Table 1). At least minimal signs of inflammation were found in 95% of samples in this group.

The results are summarized in Table 4.

No signs of inflammation were found in the control group. 

### 3.3. Correlation of Histology and Detected Compounds

Detected compounds in correlation to histology (group A—severity of inflammation 0–2-55.0% (22/40) of samples; group B—severity of inflammation 3-45.0% (18/40) of samples respectively) are summarized in Table 5.

Examples of histological findings in the samples of nasal mucosa are provided in Figure 6, Figure 7, Figure 8 and Figure 9.

### 3.4. Correlation of Smoking Status and Detected Compounds

As to the smoking status, 22.5% (9/40) and 30.0% (3/10) of samples in the CRS group and control group, respectively, were obtained from smokers. All the samples obtained from smokers in the CRS group belonged to the group A with no or mild histological signs of inflammation. 

Detected compounds in the CRS group correlated to the smoking status are summarized in Table 6.

In the control group, both samples in which TiO_2_ and amorphous carbon were found were obtained from non-smokers. No compounds were detected in the smoker subgroup of the control group. 

### 3.5. Correlation of Occupational History and Detected Compounds

Detected compounds in the CRS group (group M/O respectively) are summarized in Table 7.

In the control group, the sample in which TiO_2_ was found was obtained from a “manual” worker, while the sample with amorphous carbon detected was obtained from a “office” worker.

## 4. Discussion

The aim of the presented study was to detect MPs and NPs in the nasal mucosa tissue obtained from patients diagnosed with CRS, correlate their presence with the histological findings in the mucosa and to compare their presence and molecular composition with the control group without history of CRS. Since a vast variety of these compounds were found in the CRS group while their presence in the control group was rare, the presented results suggest an association between the diagnosis of CRS and observed particles. 

Although the authors’ previous pilot study has proposed a somewhat limited possibility of quantification of solid particles in the mucosa and their localization, with the ongoing examination of the samples these methods have proven to be immensely time-consuming, and the obtained data of low informative value [16]. Therefore, the authors focused on the sole examination of the presence of the particles to bring more detailed information about their composition in larger set of samples compared to a representative control group. Although the authors are aware that the lack of quantification of the particles poses a limitation to this study, in their opinion the presented results, especially the striking contrast between the abundant presence of particles in the CRS group and virtual lack of particles in the control group, can still contribute to better understanding of CRS pathogenesis, as will be discussed further on. 

As to the composition of the particles found in the tissue samples, the compounds that were found in the tissues the most frequently were either carbon compounds (graphite, amorphous carbon, calcite and ankerite) or titanium dioxide (in the crystalic form of anatase and rutile). Both carbon and titanium compounds have been studied extensively in the past 2 decades for their cytotoxicity, genotoxicity and pro-inflammatory properties, predominantly in vitro and in animal models [9,10,11,23,24,25,26,27,28]. The results of this study are therefore in accordance with the general direction of the field of nanotoxicology with numerous studies focusing on possible health implications of manufactured carbon- and titanium-based nanomaterials (carbon nanotubes, printer ink, pigments, cosmetics compounds etc.) [12].

Iron-based MPs and NPs of exogenous origin have also been previously found in tissue samples by Raman microspectroscopy [6,8,16]. Iron oxide dust and fumes are known to cause harm to respiratory tract (pneumosiderosis) which has been documented in welders occupationally exposed to welding fumes and are also emitted by fossil fuel combustion into the environment [6,29].

BaSO_4_ (barite), Al (aluminum) and Si (silicon) were detected sporadically; these compounds are frequently found in road traffic and vehicle non-exhaust emissions (wear debris of brake materials) [30].

Overall, the composition of the compounds detected in this study can confirm the premise that the solid anorganic particles found in nasal mucosa are of exogenous origin. Although one of the limitations of the study is the lack of a control group consisting of residents from different regions for the purpose of comparison of the particle composition according to the specific living environment, virtually all the compounds detected in the tissue samples are considered either generally ubiquitous air pollutants and/or are linked to smoking or occupational exposure, as discussed above (none of the detected compounds can be considered “region-specific”). 

The striking difference between the presence of solid particles in the CRS and the control group is possibly the most remarkable result obtained in this study. Epithelial barrier dysfunction and its increased permeability is one of the pathogenetic mechanisms in chronic mucosal inflammation and has been studied in chronic rhinosinusitis, as well [12,31,32]. This could be the possible explanation to our findings—in healthy mucosa, the deposition of solid particles is prevented by functional epithelial barrier mechanisms (mucocilliary transport, intercellular junctions, intact basement membrane, etc.), while in CRS, the mucosal epithelium is permeable to airborne particles. Given the cytotoxic and pro-inflammatory properties of said particles, their presence in the mucosa can possibly promote and maintain further chronic inflammatory processes and contribute to refractory CRS symptoms. 

Histological signs of severity of inflammation in the tissue samples were subjectively assessed by a single pathologist. A standard histologic optic-microscopy examination was performed in all the samples of mucosa; no signs of inflammation were found in the control group samples while at least mild signs of inflammation, from epithelial hyperplasia to definite signs of chronic inflammation (dense inflammatory cellularization) were found in 95% of CRS group samples. This is in correlation with the presence of solid particles in 90% of the CRS group samples and the absence of these particles in healthy mucosa in 80% of the control group samples. 

While semiquantitative histological grading scales have been proposed in different inflammatory processes, for example in ulcerative colitis [33], no similar scale exists in case of CRS. Therefore, for the purpose of correlation of presence of particles and histologic findings, the authors divided the CRS group into two subgroups: group A—no or mild signs of inflammation; and group B—severe inflammation. Against the authors’ expectations, significantly larger variety of compounds were found in group A compared to in group B (Table 1 and Table 5). 

The authors are aware that this evaluation has its limits (subjective assessment, examination of a single section of mucosa that might not be representative, limited possibility of quantification of the detected compounds) and that further research is needed to determine the relationship between severity of inflammation and presence (and possibly quantity) of solid particles. 

The fact that no definite histological signs of chronic inflammation were found in any of the samples obtained from patients with diagnosed CRS, defined as a chronic inflammatory disease, is in accordance with the concept of CRS as a clinical diagnosis that should be treated according to the endoscopic findings and patients’ complaints [20].

Smoking status and occupational history indicate increased exposition of smokers and certain workers to airborne particles. In the study, significant association of smoking and high-risk (“manual”) occupation with presence of carbon compounds was found, while these two characteristics are inseparable (all the smokers in the CRS group were “manual” workers) because of clear association of daily smoking and social class [12,34].

## 5. Conclusions

The results of this study are in accordance with the premise that exogenous airborne MPs and NPs, predominantly carbon and titanium compounds, interact with the nasal mucosa and possibly deposit in it in case the epithelial barrier is compromised in CRS. In control samples, i.e., in healthy mucosa, as confirmed histologically, these particles were virtually absent. Thus, these results suggest association between the diagnosis of CRS and observed particles. 

Presence of carbon compounds seem to be associated with smoking and occupations in high risk of air pollutant exposure.

It is unclear whether presence of MPs and NPs contributes to the severity of inflammation; in this study, evaluation of histological findings in correlation with the presence of solid particles in the mucosa was inconclusive (however, this is in accordance with the concept of CRS being a clinical diagnosis based on endoscopic examination of the nasal cavity and patients’ complaints).

Although it can be concluded that the presence of these compounds may be linked to CRS, further research is needed to determine if the link is causal or that presence of solid particles in the mucosa is a result of pre-existing impairment and increased permeability of the epithelial barrier in chronic inflammation. 

## Figures and Tables

**Figure 1 ijerph-19-07269-f001:**
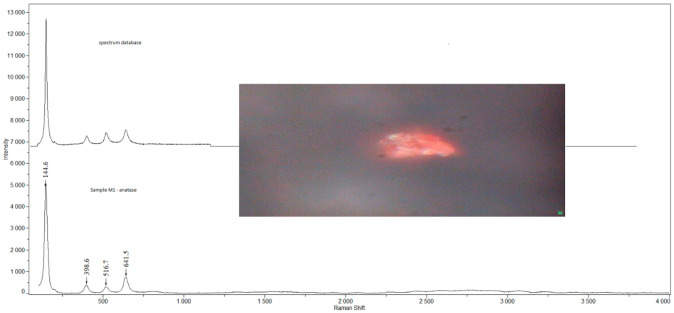
Raman spectra and electron microscopy image of anatase found in sample M1.

**Figure 2 ijerph-19-07269-f002:**
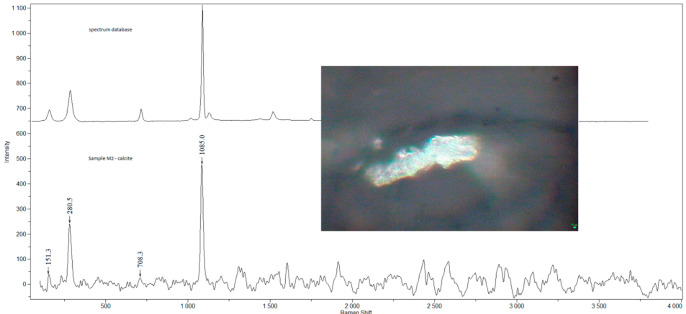
Raman spectra and electron microscopy image of calcite found in sample M2.

**Figure 3 ijerph-19-07269-f003:**
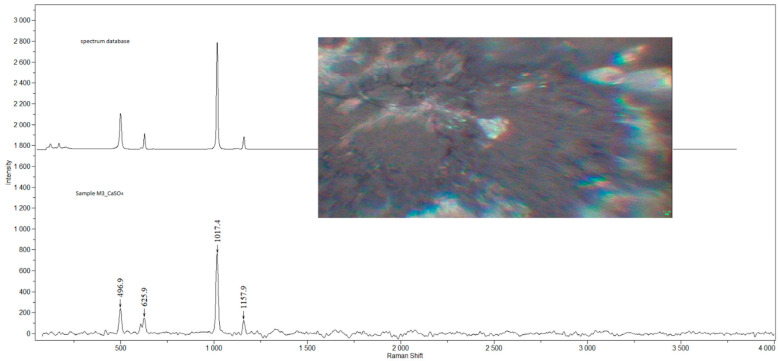
Raman spectra and electron microscopy image of CaSO_4_ found in sample M3.

**Figure 4 ijerph-19-07269-f004:**
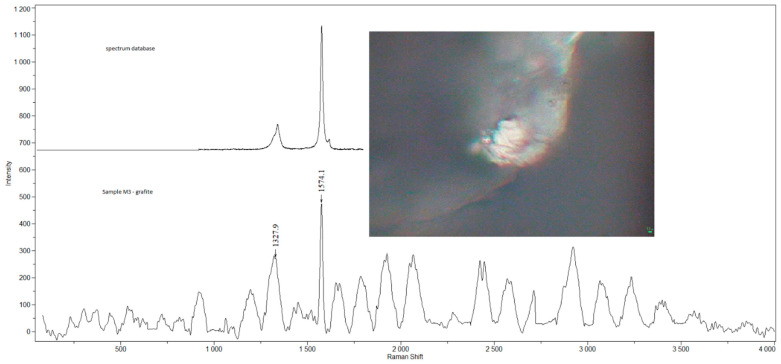
Raman spectra and electron microscopy image of graphite found in sample M3.

**Figure 5 ijerph-19-07269-f005:**
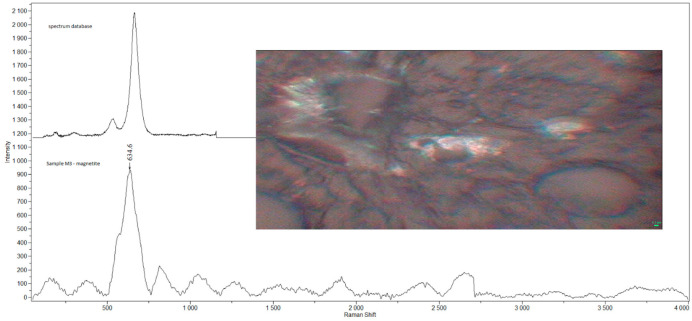
Raman spectra and electron microscopy image of magnetite found in sample M3.

**Figure 6 ijerph-19-07269-f006:**
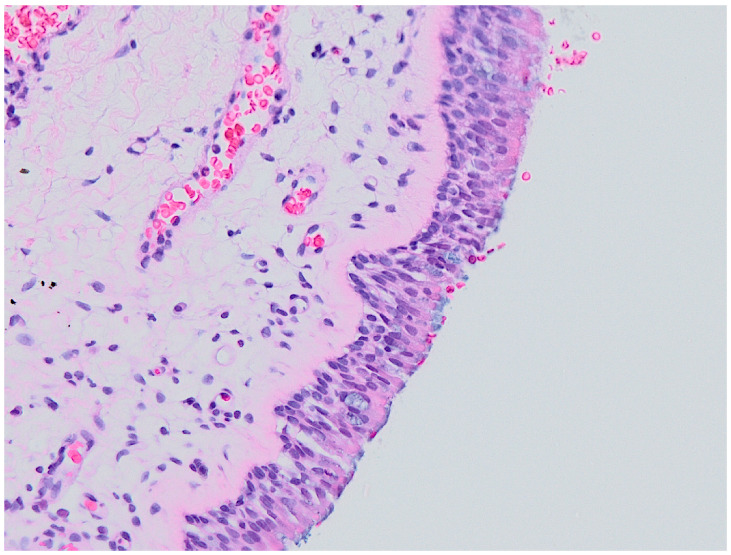
Normal histology (no inflammation, severity of inflammation grade 0). Optic microscopy, magnification 200×.

**Figure 7 ijerph-19-07269-f007:**
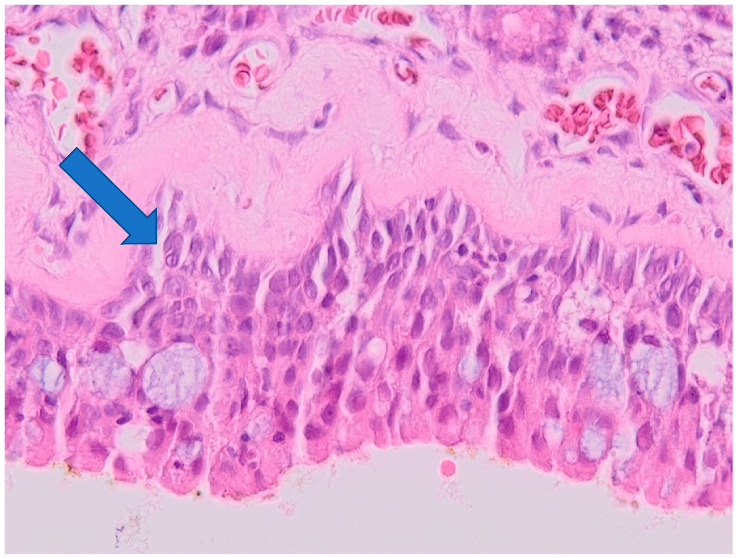
Epithelial hyperplasia (severity of inflammation grade 1). Arrow indicates hyperplastic epithelium. Optic microscopy, magnification 200×.

**Figure 8 ijerph-19-07269-f008:**
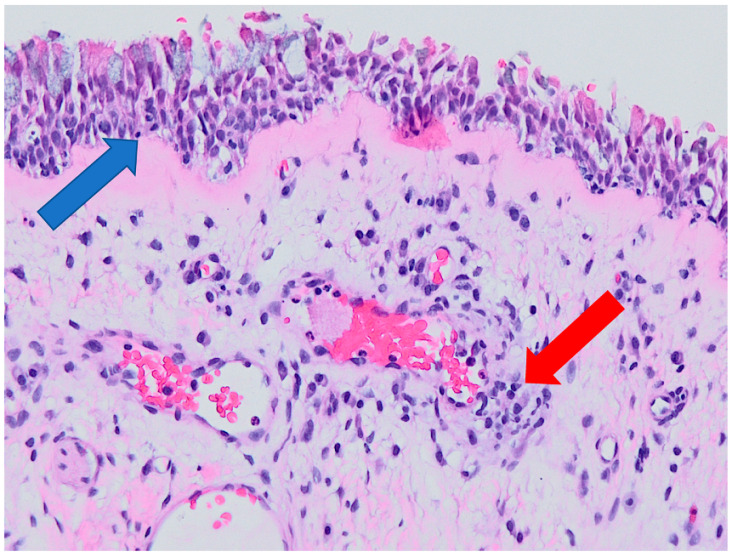
Epithelial hyperplasia with mild signs of inflammation (severity of inflammation grade 2). Arrows indicate hyperplastic epithelium (blue) and focal inflammatory cellularization (red). Optic microscopy, magnification 200×.

**Figure 9 ijerph-19-07269-f009:**
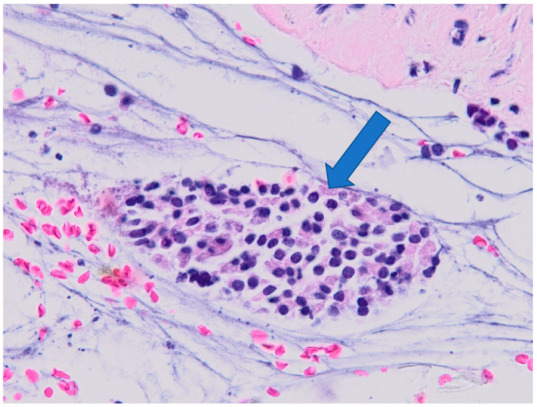
Chronic inflammation of the submucosa (severity of inflammation grade 3). Arrow indicates a large cluster of inflammatory cellularization. Optic microscopy, magnification 400×.

**Table 1 ijerph-19-07269-t001:** CRS group.

Sample	Sex	Age	Smoking Status	Occupation	Detected Compounds	Histology (Severity of Inflammation)
M11	m	42	Y	builder (M)	GR	0
M32	m	31	N	administrator (O)	AC, Al comp., ankerite	0
M1	m	78	N	welder (M)	AC, ankerite, CaCO_3_, GR, TiO_2_-A	1
M2	m	38	N	programmer (O)	CaCO_3_, Fe_3_O_4_, TiO_2_-A, TiO_2_-R	1
M3	m	38	N	welder (M)	AC, CaSO_4_, Fe_3_O_4_, GR, TiO_2_-A	1
M4	m	65	Y	policeman (M)	AC, ankerite, GR, TiO_2_-A, TiO_2_-R	1
M14	m	32	N	policeman (M)	AC, Fe_2_O_3_, GR, TiO_2_-A	1
M16	m	44	Y	driver (M)	CaCO_3_, GR, TiO_2_-A, TiO_2_-R	1
M17	f	42	N	warehouse keeper (M)	ankerite, BaSO_4_, TiO_2_-A	1
M18	m	34	Y	carrier (M)	AC, CaCO_3_, GR, SiO_2_, TiO_2_-A	1
M20	m	35	Y	tinsmith (M)	ankerite, CaCO_3_, GR, SiO_2_	1
M33	m	43	N	machinist (M)	AC, GR, TiO_2_-A	1
M35	f	65	N	manager (O)	CaCO_3_, TiO_2_-R	1
M40	m	54	N	bailiff (O)	-	1
M8	m	44	Y	labourer (M)	CaCO_3_, GR	2
M10	m	37	N	manager (O)	GR, TiO_2_-A	2
M21	m	31	N	labourer (M)	ankerite, GR	2
M2	m	28	N	rolling mill operator (M)	TiO_2_-A	2
M25	m	44	N	executive director (O)	AC, Si comp., TiO_2_-A, TiO_2_-R	2
M28	m	42	Y	policeman (M)	AC, ankerite, BaSO_4_, GR	2
M29	m	25	Y	unemployed (O)	AC, GR, TiO_2_-A	2
M30	m	39	Y	welder (M)	AC, Al comp., (Ca, Mg)CO_3_	2
M5	f	26	N	student (O)	Fe_3_O_4_, GR	3
M6	f	44	N	labourer (M)	AC, CaCO_3_	3
M7	m	28	N	student (O)	GR	3
M9	f	58	N	artist (O)	GR, TiO_2_-A	3
M12	f	45	N	manager (O)	GR, TiO_2_-A	3
M13	m	40	N	clerk (O)	Fe_2_O_3_, TiO_2_-A	3
M15	f	53	N	shop assistant (M)	ankerite, GR, TiO_2_-A	3
M19	m	48	N	waiter (M)	ankerite, CaCO_3_, Fe_2_O_3_, SiO_2_, TiO_2_-A	3
M23	m	28	N	operator (M)	TiO_2_-A	3
M24	m	54	N	policeman (M)	AC, GR	3
M26	m	42	N	train dispatcher (M)	AC, CaCO_3_, GR	3
M27	m	25	N	student (O)	AC	3
M31	f	20	N	student (O)	GR, TiO_2_-A, TiO_2_-R	3
M34	f	41	N	seamstress (M)	GR, TiO_2_-A	3
M36	m	36	N	IT technician (O)	GR, TiO_2_-A	3
M37	m	44	N	production supervisor (M)	-	3
M38	m	34	N	electrotechnician (M)	-	3
M39	m	55	N	businessman (O)	-	3

Table legend: m—male; f—female; Y—yes; N—no; M—manual worker; O—office worker; GR—graphite; AC—amorphous carbon; comp.—composite; TiO_2_-A—anatase; TiO_2_-R—rutile. Histology (level of inflammation): 0—normal histology (no inflammation); 1—epithelial hyperplasia; 2—epithelial hyperplasia with mild signs of inflammation in submucosal space (characterized as “focal”; “discrete” or “minimal” inflammatory cellularization); 3—chronic inflammation.

**Table 2 ijerph-19-07269-t002:** Control group.

Sample	Sex	Age	Smoking Status	Occupation	Detected Compounds
R1	m	71	N	M	-
R5	m	57	Y	M	-
R6	m	77	N	M	TiO_2_
R7	f	78	N	O	AC
R8	m	44	N	M	-
R9	m	67	Y	M	-
R10	m	37	Y	M	-
R11	f	87	N	O	-
R12	f	84	N	O	
R13	f	84	N	O	-

Table legend: m—male; f—female; Y—yes; N—no; M—manual worker; O—office worker; AC—amorphous carbon.

**Table 3 ijerph-19-07269-t003:** Main detected compounds in the CRS group tissue samples (total of 40 samples).

Detected Compound	Number of Samples/40	Percent of Samples
Graphite TiO_2_ amorphous carbon	24	60.0%
	23	57.5%
	15	37.5%
CaCO_3_ Ca(Fe, Mg, Mn)(CO_3_)_2_	10	25.0%
	9	22.5 %
iron compounds	6	15.0%

**Table 4 ijerph-19-07269-t004:** Severity of inflammation in the CRS group (total of 40 samples).

Severity of Inflammation	Number of Samples/40	Percent of Samples
0 (no inflammation) 1 (epithelial hyperplasia) 2 (mild signs of inflammation)	2	5.0%
	13	32.5%
	7	17.5%
3 (chronic inflammation)	18	45.0%

**Table 5 ijerph-19-07269-t005:** Detected compounds in correlation to severity of inflammation in the CRS group (group A—severity of inflammation 0–2; group B—severity of inflammation 3).

Detected Compound	Group A/22 Samples		Group B/18 Samples	
	Number of Samples	Percent of Samples	Number of Samples	Percent of Samples
graphite	14	63.0%	9	50.0%
TiO_2_	14	63.0%	9	50.0%
amorphous carbon	7	31.8%	4	22.2%
CaCO_3_	7	31.8%	3	16.7%
Ca(Fe, Mg, Mn)(CO_3_)_2_	7	31.8%	2	11.1%
iron compounds	3	13.6%	3	16.7%

**Table 6 ijerph-19-07269-t006:** Detected compounds in correlation to smoking status in the CRS group.

Detected Compound	Smokers/9 Samples		Non-Smokers/31 Samples	
	Number of Samples	Percent of Samples	Number of Samples	Percent of Samples
graphite	8	88.9%	16	51.6%
TiO_2_	4	44.4%	19	61.3%
amorphous carbon	5	55.6%	10	32.3%
CaCO_3_	4	44.4%	6	19.4%
Ca(Fe, Mg, Mn)(CO_3_)_2_	3	33.3%	6	19.4%
iron compounds	0	0.0%	6	19.4%

**Table 7 ijerph-19-07269-t007:** Detected compounds in correlation to occupational history in the CRS group.

Detected Compound	Manual Workers/22 Samples		Office Workers/18 Samples	
	Number of Samples	Percent of Samples	Number of Samples	Percent of Samples
graphite	17	77.3%	7	38.9%
TiO_2_	13	59.1%	10	55.5%
amorphous carbon	11	50.0%	4	22.2%
CaCO_3_	8	36.4%	2	11.1%
Ca(Fe, Mg, Mn)(CO_3_)_2_	8	36.4%	1	5.6%
iron compounds	3	14.6%	2	11.1%

## Data Availability

The data presented in this study are available on request from the corresponding author.

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
