# Peer review of "Solid Anorganic Particles and Chronic Rhinosinusitis: A Histopathology Study"

_ijerph, 2022, doi:10.3390/ijerph19127269_

Round 1

Reviewer 1 Report

The work is described "Solid Anorganic Particles and Chronic Rhinosinusitis: A Histopathology Study  "by using Raman microspectroscopy and histological tests. It is very interesting work. However, there are some suggestions listed below. 

1. The inclusion and exclusion criteria of study should be described more clearly. Notably, there are some different in two groups (ex; age/occupation in control group/CRS groups)

2. The major issue of this study is sample size. There are some factors was suggested to add in the study setting (ex: living environment/air conditions/immune status/). 

3. In the section of discussion, please describe the limitation of this study.

Author Response

Review 1:

  1. The inclusion and exclusion criteria of study should be described more clearly. Notably, there

are some different in two groups (ex; age/occupation in control group/CRS groups)

We changed this section of the paper and discuss the inclusion and exclusion criteria in both groups in further detail.

  1. The major issue of this study is sample size. There are some factors was suggested to add in

the study setting (ex: living environment/air conditions/immune status/).

All the patients‘ and cadaveric donors‘ living environment was similar, as the residence of all of them was in the same region of the Czech Republic (Moravian-Silesian region). We are aware that air pollution specific for a given region may influence the results, however the composition of the detected particles correlates with the composition of the general ubiquitous air pollution and or/specific occupational exposure which is discussed in the paper (no “surprise” compound presence of which could have been explained only by the specific living environment was found). Moreover, we believe that the most remarkable result of the study (i.e. abundant presence of particles in CRS group and their virtual absence in healthy control group that can be explained by the increased mucosal permeability in chronic inflammation), if set in a different region with different type of air pollution, would probably not be any different. Since we did not enroll any patients from other regions, we cannot discuss any possible differences in the composition of the particles as to the living environment.

All the patients enrolled in the study were immunocompetent and apt of surgery in general anaesthesia.

  1. In the section of discussion, please describe the limitation of this study.

The limitations of the study (mainly lack of quantification of the particles and subjective assessment of the histologic findings) are discussed in the paper. We also newly discuss the limitation as to the living environment specificity, as suggested.

Reviewer 2 Report

Dear Authors, 

congratulationd for the study, 

However results must be presented better: Figures should be cited in the text properly and histology picture should be included as well as with arrows indicating the inflamation etc, 

One question: if the study finished in 2015 why did You submitted in 2022?

Maybe the data can be outdated?

Author Response

Figures should be cited in the text properly and histology picture should be included as well as with arrows indicating the inflamation etc,

We made all the necessary adjustments, included the figures into the text and added the arrows indicating the inflammation.

If the study finished in 2015 why did You submitted in 2022? Maybe the data can be outdated?

The acquisition of the samples ended in 2015, however our authors’ collective was able to finalize the results only in 2021, specifically the RMS examination of the tissue samples because this method is rather time consuming. To our best knowledge, no similar histopathology study has been published so far. We believe that this work presents a new unique set of results that is in no way outdated.

Reviewer 3 Report

I have carefully read the study entitled as "Solid Anorganic Particles and Chronic Rhinosinusitis: A Histo-2 pathology Study" by Cabalova et al. It is really an interesting paper. but I have few commnets:

1) Minor spelling and grammar check is necessary.

2) It would be good if they show the general characteristics of the patients as Table 1. It may include gender, age, BMI, smoking history..

3) Smoking or other exposure history may be important to differentiate the patients net exposure. The smoking history classification as well as pack year information may be helpful.

Author Response

It would be good if they show the general characteristics of the patients as Table 1. It may

include gender, age, BMI, smoking history.

Thank you for the recommendation. We believe that the characteristics of the patients from the CRS group as shown in Table 1 (gender, age, smoking and occupational history) provide clear information about the study subjects, therefore we decided not to list them in a separate table. In our opinion, BMI of the patients is not very relevant to the results, therefore we did not include this data.

Smoking or other exposure history may be important to differentiate the patients net exposure.

The smoking history classification as well as pack year information may be helpful.

Thank you for the suggestion. Unfortunately, we cannot provide data on net exposure to smoking because such data was not collected in the study. However, although the results indicate quite clear correlation of smoking and detected carbon-based compounds, their quantity was not evaluated in the study and therefore could not be correlated to pack/year exposure.

Round 2

Reviewer 1 Report

I appreciate the efforts that the authors have made in response to my concerns. However, the related factors was suggested to validate by using molecular/serological/epidemiological methods. 

Reviewer 2 Report

manuscript can be now accepted 

This manuscript is a resubmission of an earlier submission. The following is a list of the peer review reports and author responses from that submission.

Round 1

Reviewer 1 Report

The authors of this paper present a strategy to detect micron and submicron particles in the nasal mucosa tissue obtained from patients diagnosed with chronic rhinosinusitis and in a control group by means of Raman spectroscopy and to establish an association between the diagnostic and the presence of particles. Compared with their previous studies, it is not clearly stated what is the novelty of the present study (ref 12, 14).

It is not clear what kind of particles was found. In the title and introduction are mentioned nanosized particles, but the discussion section the micron and submicron particles are mentioned. The author should add some supplementary data (e.g. electron microscopy images) to sustain their affirmation. Also, they use Raman spectroscopy to analyze the chemical nature of various particles and they should present some corresponding spectra (either in manuscript or as supplementary material).

Best regards,

R

Reviewer 2 Report

The authors focused on their detection in nasal mucosa of patients with chronic rhinosinusitis and the results were compared to the findings in mucosa obtained from cadavers with no history of chronic rhinosinusitis. The composition of the particles was compared to the histologic findings and solid particles were found in 90% of tissue samples in the group with chronic rhinosinusitis. This manuscript is containing some informative results. But, as the authors point out, there are important defects in publishing.

1. Major point

The lack of quantification of the particles is very serious defect in this manuscript. By this defect, we can’t judge including its possibility whether the particles effects for CRS or not. It is strongly desired the quantification for publishing as a scientific manuscript. Quantitative data of particles in urine and blood may also be helpful.

2. Lines 204-, the histological pictures for typical findings are desired.

3. Lines 204-, More detailed pathological search is needed, types and numbers of inflammatory cells, etc.